# New seasonal pattern of pollution emerges from changing North American wildfires

Rebecca R. Buchholz [1✉], Mijeong Park[1], Helen M. Worden [1✉], Wenfu Tang[1], David P. Edwards [1], Benjamin Gaubert [1], Merritt N. Deeter [1], Thomas Sullivan[2], Muye Ru[3], Mian Chin [4], Robert C. Levy [4], Bo Zheng [5] & Sheryl Magzamen [6]

Rising emissions from wildfires over recent decades in the Pacific Northwest are known to counteract the reductions in human-produced aerosol pollution over North America. Since amplified Pacific Northwest wildfires are predicted under accelerating climate change, it is essential to understand both local and transported contributions to air pollution in North America. Here, we find corresponding increases for carbon monoxide emitted from the Pacific Northwest wildfires and observe significant impacts on both local and down-wind air pollution. Between 2002 and 2018, the Pacific Northwest atmospheric carbon monoxide abundance increased in August, while other months showed decreasing carbon monoxide, so modifying the seasonal pattern. These seasonal pattern changes extend over large regions of North America, to the Central USA and Northeast North America regions, indicating that transported wildfire pollution could potentially impact the health of millions of people.

[1] Atmospheric Chemistry Observations & Modeling Laboratory, National Center for Atmospheric Research, Boulder, CO, USA. [2] University of Colorado, Boulder, CO, USA. [3] Columbia University, New York, NY, USA. [4] NASA Goddard Space Flight Center, Greenbelt, MD, USA. [5] Institute of Environment and Ecology, Tsinghua Shenzhen International Graduate School, Tsinghua University, Shenzhen 518055, China. [6] Colorado State University, Fort Collins, CO, USA. ✉email: buchholz@ucar.edu; hmw@ucar.edu

Wildfires in Northwest America have been increasing over recent decades[1–3]. Both the extent and duration of the wildfire season have expanded[2,4], and are linked to changing climate through drought severity and fuel dryness[1,5]. Models predict that future climate change will further increase the probability of wildfire in this region by creating more frequent hot and dry conditions[5–7]. Humans also impact the occurrence of wildfires through land use change[8], increasing ignitions[9], and land management policies such as fire suppression and prescribed burning[10]. Wildfire smoke exposure is detrimental to human health for communities close to wildfires, as well as for downwind communities that experience transported pollution[11]. Thus, understanding both the local and transported contributions to poor air quality from wildfire pollution is critical for optimizing responses in the health system and mitigating future adverse health impacts.

Increasing wildfires have already been linked to degrading air quality in the USA with respect to the fine particulate matter as observed in the form of organic aerosols[12]. North American anthropogenic emissions that result in aerosols have been decreasing, improving air quality for the Eastern States[13]. At the same time, wildfire pollution is driving an upward trend in aerosols for the Northwest[12,13]. The influence of Pacific Northwest (PNW) wildfires on other atmospheric trace gas and aerosol pollutants, as well as the down-wind impacts on air quality and human health, requires further investigation.

Along with aerosols, carbon monoxide (CO) is emitted from fires during incomplete combustion[14,15]. With an atmospheric lifetime ranging from weeks to months, CO is valuable for tracking the atmospheric transport of large sources of pollution, such as from wildfires[16]. The atmospheric evolution of wildfire-emitted CO is also important to help understand distributions of other atmospheric species related to wildfire, such as the health-relevant tropospheric ozone[17] that is photochemically produced in the wildfire plume. Globally, atmospheric CO has been decreasing for the last two decades, primarily due to improvements in the combustion efficiency of anthropogenic source processes[18–23], and to a global decline in tropical fires[8]. However, a recent slow-down in the decreasing trend of Northern Hemsiphere CO, particularly in the summer months, may be linked to climate-driven increases in high-latitude wildfires such as those observed in the North American PNW[23].

In this work, we examine the atmospheric impact of PNW wildfire emissions between 2002 and 2018 for three North American regions with the aim to quantify near- and far-field impacts on air quality. We use the long record of satellite-measured CO from the Measurements of Pollution In The Troposphere (MOPITT)[24] instrument to investigate how CO abundance during the months of peak burning in the PNW differs with other months. Seasonal pattern changes in regional CO abundance are also used to highlight the extent of local and downwind impacts from increasing wildfire emissions in the PNW. Different fire and anthropogenic emission inventories are used to support that PNW wildfire is driving the observed seasonal pattern changes. Our study suggests that smoke-related health impacts that are predicted to worsen with climate change may already be emerging.

## Results

### Upward trend in August CO for the Pacific Northwest.
Atmospheric background CO concentrations are declining globally at an average rate of −0.50% per year between 2002 and 2018[23]. However, we find an upward trend in August CO over large regions of North America (Fig. 1a). In other months, North American CO is declining in line with the global background

trend (Supplementary Fig. 1). The strongest increasing trend in August CO observed by MOPITT is focused within the PNW region (38°–57°N, 127°–110°W), but the upward trend extends through Central USA to the Northeast of the continent. The dominant increases in CO focused in the PNW suggests that regionally, local emissions are counteracting the globally-observed downward trend in CO. To the west of the PNW over the Pacific Ocean, negative trends in CO are observed in August. Recent work[23] identified downward trends in the Northern Hemisphere background CO in all months, as well as strong downward trends in CO over Northeast China, suggesting transported pollution into North America via westerly flow does not play a large role in the August positive CO trend. Globally, this CO increase is unique for the North American continent during August, and has potential hemispheric influence (Supplementary Fig. 2). As increased emissions are taken up in the prevailing westerlies, the upward trend in CO extends over the transatlantic transport pathway to Europe[25,26].

A similar spatial pattern to CO is seen in the trend analysis of August aerosol optical depth (AOD) measured from space with the Moderate Resolution Imaging Spectroradiometer (MODIS)[27] (Fig. 1b). In the PNW, aerosols are known to be increasing due to increasing smoke from summertime wildfires[12,13]. This is in contrast to the Southeast USA, where aerosols are decreasing due to air quality management policy restrictions on anthropogenic emissions[12] (also apparent in Fig. 1b). While the Northeast contains a high human population (~72 million[28]) and would therefore be expected to show aerosol decreases due to anthropogenic pollution reductions, Fig. 1b shows no significant downward trends for this region. This suggests that transported wildfire pollution may be offsetting the gains in improving air quality from anthropogenic reductions over the Northeast.

We consider CO and AOD in three regions chosen to investigate local and transported impacts on pollution loadings from the PNW wildfires: the local PNW, a Central USA region, and the further downwind Northeast region of North America (regions defined in Fig. 1). A time series of regional monthly average MOPITT CO in the PNW is displayed as column average volume mixing ratio (VMR) in Fig. 2. We compare this to the regional monthly average aerosols using AOD (Fig. 2). After 2011, an August CO peak emerges. This coincides with a strengthening of the August AOD peak due to the positive trend in AOD that has been attributed to increased wildfire aerosol[12,13]. The spatial and temporal co-evolution of AOD and CO in August indicates a similar local source, providing empirical support that the emergence of the August CO peak is due to wildfires. These secondary CO peaks also coincide with peak burning in the PNW, as described by MODIS fire count and burned area (Supplementary Fig. 4), further supporting a link between wildfires and the CO August peak. The magnitude of peak PNW burning is generally larger in 2012–2018 compared to 2002–2011. All three regions, PNW, Central USA, and the Northeast, exhibit the emergence of an August CO peak in later years (Supplementary Fig. 5).

### Changing North American seasonal patterns.
Based on the emergence of the CO peak after 2011, we separate the time series into two time periods, 2002–2011 and 2012–2018, to investigate the average seasonal cycles. In addition to CO column-average VMR and AOD, MOPITT surface layer CO is also investigated in order to identify potential air quality impacts for humans. The average Northern Hemisphere background trend in atmospheric CO was removed (−0.57% per year[23]), prior to calculating the CO seasonal patterns.

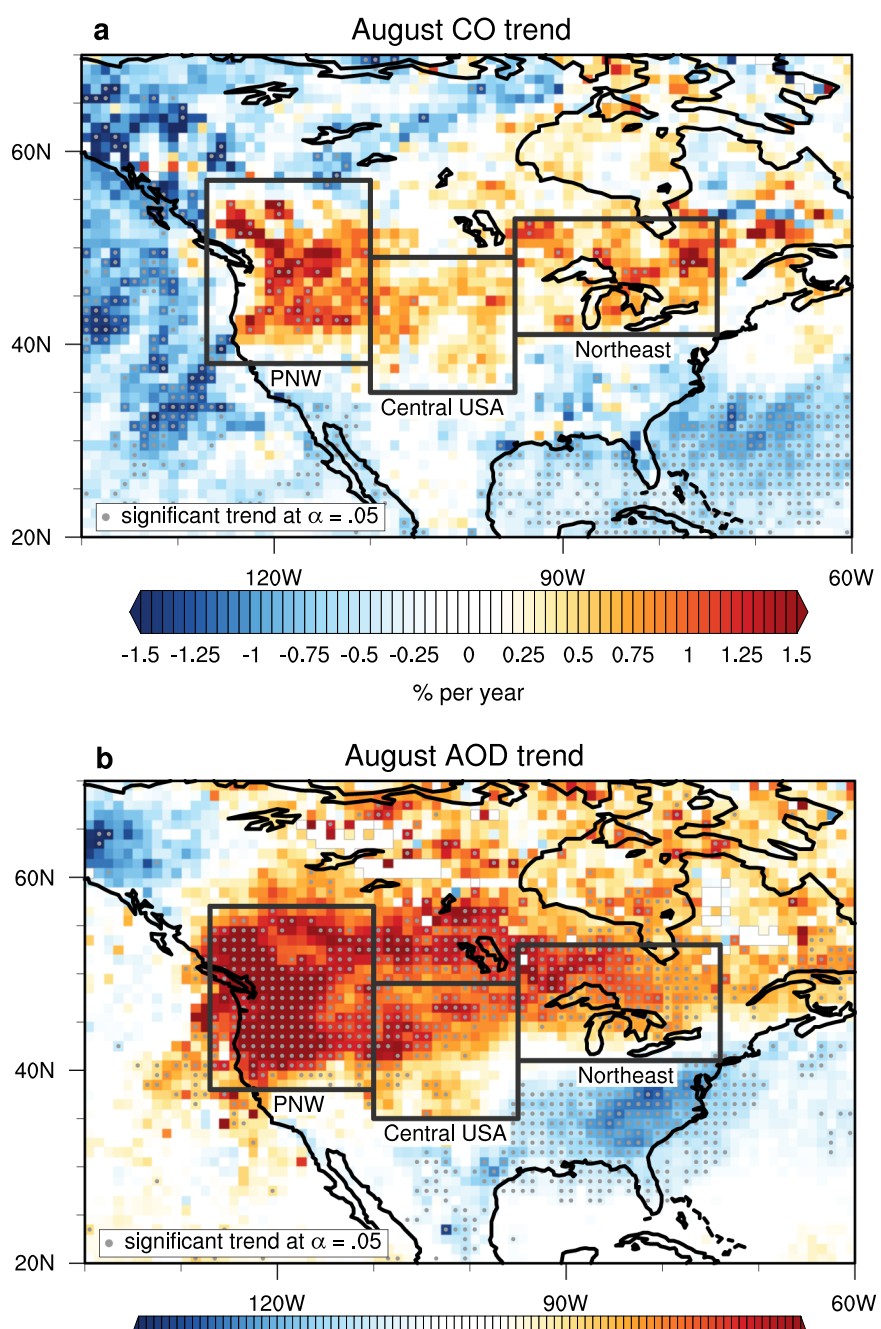

**Fig. 1 August trend analysis of atmospheric composition.** August trend (2002–2018) in satellite-measured atmospheric composition is shown in a 1×1 degree grid. **a** Column carbon monoxide (CO) trends (n = 16) and **b** aerosol optical depth (AOD) trends (n = 17). Gray dots indicate significant non-zero trends, α = 0.05. Defined regions are outlined in black: Pacific Northwest (PNW, 38°–57°N, 127°–110°W), Central USA (35°–49°N, 110°–95°W), and the Northeast (41°–53°N, 95°–74°W). White areas outlined in gray denote missing data.

In all three regions and for both time periods, CO loading shows a photochemically-driven maximum during Northern Hemisphere spring, in April (Fig. 3a–c). The CO seasonal cycle results from a combination of source and loss mechanisms, with loss dominated by reaction with the photochemically produced hydroxyl (OH) radical[18]. Due to seasonal variability in sunlight, the chemical lifetime of CO over winter is about 2 months, compared to less than a month in summer when photochemical production of OH is at a maximum[29]. Consequently, in atmospheres with well-mixed atmospheric conditions (i.e., homogeneous properties) that are distant from sources, CO accumulates over winter to peak in late winter/spring and shows a minimum in late summer. Deviations from this OH-driven seasonal cycle are caused by anomalous sources, and on a large-scale are often due to wildfires[23]. The OH-driven spring CO peak and late summer minimum is the seasonal pattern observed in column CO for all regions prior to 2011. In contrast, the recent time period (2012–2018) shows an emerging summer CO peak

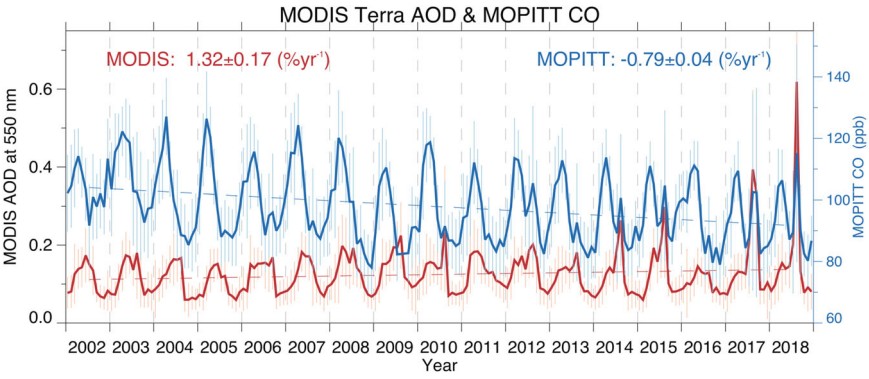

**Fig. 2 Pacific Northwest time series and trend analysis of atmospheric composition from 2002 to 2018.** We show Measurements of Pollution in the Troposphere (MOPITT) monthly average carbon monoxide (CO) as units of column average volume mixing ratio in parts per billion (ppb, blue) and Moderate Resolution Imaging Spectroradiometer (MODIS) monthly average aerosol optical depth (AOD, red). Error bars represent monthly standard deviation. Vertical gray dashed lines indicate the beginning of each year. Trends are determined using the weighted least squares estimate of the slope from data anomalies, and include the standard error of the slope estimate. For CO the mean sample size, *n*, is 1110 observations; for AOD the mean sample size, *n*, is 240 observations. Full details of all *n* values are provided in the Source Data file.

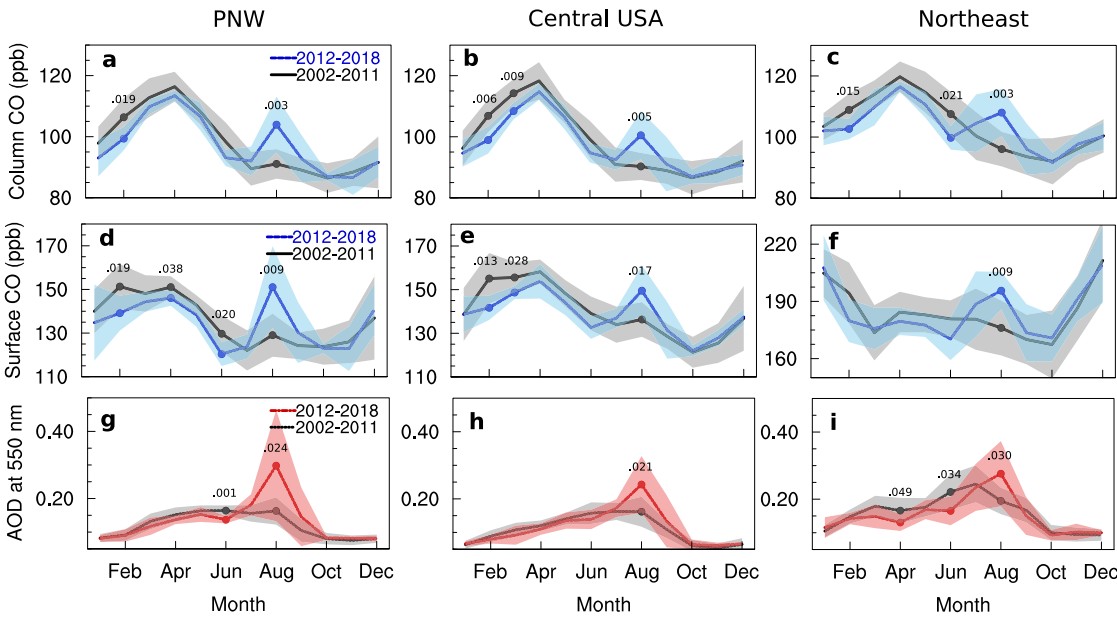

**Fig. 3 Seasonal patterns of atmospheric composition.** Regional seasonal patterns of atmospheric composition are shown for two different time periods: 2002–2011 (black line with gray shading) and 2012–2018 (blue or red). **a–c** Measurements of Pollution in the Troposphere (MOPITT) carbon monoxide (CO) as column average volume mixing ratio (VMR) in parts per billion (ppb). **d–f** MOPITT surface CO VMR. **g–i** Moderate Resolution Imaging Spectroradiometer (MODIS) aerosol optical depth (AOD). Regions are the Pacific Northwest (PNW), Central USA, and the Northeast. The shaded area shows the standard deviation and filled circles represent data means that are significantly different between the two time periods with *P* values noted for a two-tailed *t*-test. More complete statistics relevant for the August differences can be found in Table 1. The Northern Hemisphere background CO trend 2002–2018 of −0.50 ppb per year was removed prior to seasonal cycle calculation for (**a**)–(**f**).

(in August) during the expected photochemically driven minimum, and the seasonal pattern becomes bimodal. Surface layer CO seasonal cycles show a similar pattern of one major peak before 2011 and two peaks after 2011, indicating that the seasonal change is also impacting surface air quality (Fig. 3d–f). In all three regions, the August column and surface CO is significantly different between the 2012–2018 time period compared to the 2002–2011 time period (see Table 1), and indicates that the annual patterns are different between the two time periods.

The changes in the CO seasonal cycle coincide with a recent strengthening in the AOD August peak for the PNW and Central USA regions (Fig. 3g, h and Table 1). The AOD seasonal cycle is driven by aerosol processes and is generally out of phase with the background CO seasonal cycle in North America. AOD is a composite measurement of many different aerosol types, including primary and secondary sources, that are subject to different processes and have regional dependencies[30,31]. For aerosols, the major loss is via dry and wet deposition[32]. In North America, the

**Table 1 August comparison statistics between 2002–2011 and 2012–2018 mean atmospheric composition, for the three regions of interest from an independent, two-tailed *t*-test, for α = 0.05.**

|  | *P* | *t* | *df* | *d* | Mean difference | 95% CI |
|---|---|---|---|---|---|---|
| *PNW* | | | | | | |
| Column CO | 0.003 | −3.627 | 14 | 1.753 | −12.844 | −20.233; −5.454 |
| Surface CO | 0.009 | −3.038 | 14 | 1.468 | −22.005 | −37.122; −6.888 |
| AOD | 0.024 | −2.503 | 15 | 1.119 | −0.135 | −0.291; 0.021 |
| *Central USA* | | | | | | |
| Column CO | 0.005 | −3.284 | 14 | 1.593 | −10.194 | −16.632; −3.755 |
| Surface CO | 0.017 | −2.712 | 14 | 1.332 | −13.187 | −23.048; −3.326 |
| AOD | 0.021 | −2.585 | 15 | 1.200 | −0.0810 | −0.167; 0.004 |
| *Northeast* | | | | | | |
| Column CO | 0.003 | −3.581 | 14 | 1.765 | −11.948 | −18.670; −5.227 |
| Surface CO | 0.009 | −3.056 | 14 | 1.580 | −19.516 | −31.280; −7.752 |
| AOD | 0.030 | −2.400 | 15 | 1.095 | −0.0810 | −0.176; 0.014 |

*P* = P value, *t* = t statistic, *df* = degrees of freedom, *d* = Cohen's measure of sample effect size for comparing two sample means, Mean difference = 2002 to 2011−2012 to 2018, 95% CI = Confidence interval for 95%.

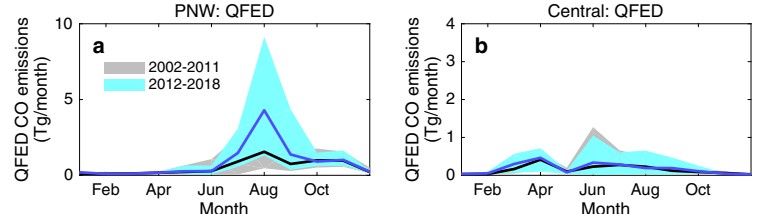
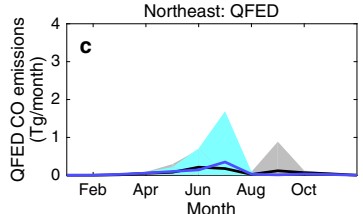

**Fig. 4 Seasonal patterns of wildfire carbon monoxide (CO) emissions.** Mean seasonal cycles for wildfire CO emissions in terragrams (Tg) in each region, **a** PNW, **b** Central USA and **c** Northeast, for each time period using the Quick Fire Emissions Dataset (QFED) inventory. Black lines follow the averaged monthly mean for 2002–2011, blue lines for 2012–2018. Shading represents the range in inter-annual variability for each month.

seasonal cycle is often dominated by secondary aerosol production that occurs through gas-phase reactions with OH (e.g., sulfates or organic species), which peaks in summer due to maximum photochemical production of OH[30,33], as well as through aqueous processing[30]. Together, these processes combine to create a broad summer season AOD maximum, as seen in the PNW and Central USA during the first time period 2002–2011. For 2012–2018, the PNW August AOD peak value is significantly larger than that for the 2002–2011 time period, showing a large effect with the mean August AOD approximately doubling between the two time periods, suggesting the emergence of a specific source—in this case, wildfires. The Central USA region also shows significantly higher August AOD for the later time period, although the difference is not as dramatic as for the PNW. In comparison, the Northeast region peak month appears to be moving from July in early years towards August for later years, with significantly higher AOD in August for 2012–2018 compared to 2002–2011. AOD is a column measurement that measures a response to the sum of aerosols throughout the atmosphere. Thus, we find that CO is a valuable tracer to examine the potential impact of PNW wildfires on Northeast air quality, because we are able to detect significant changes in the surface layer.

Our hypothesis that PNW wildfires are the driver of seasonal pattern changes in CO is supported by four different global fire emission inventories: the Fire INventory from NCAR (FINN)[34], the Global Fire Emissions Database (GFED)[35], NASA's Quick Fire Emissions Dataset (QFED)[36] and the Zheng reanalysis product[37]. All four inventories consistently reveal peak fire emissions of CO occur in the PNW during August. The PNW fire emissions are an order of magnitude higher than emissions from fires in Central USA and the Northeast (Fig. 4 and Supplementary Fig. 6). Inventories also show enhanced August fire CO in the PNW for 2012–2018 compared to 2002–2011 (Fig. 5). In comparison, fire emissions summed over Central USA and the Northeast do not peak during August and have no differences between time periods, indicating that fires local to these regions are not driving the changes in observed CO. Additionally, two anthropogenic emission inventories (Copernicus Atmosphere Monitoring Service Global Anthropogenic emissions, CAMS-GLOB-ANT[38], and Zheng reanalysis[37]) show that CO emissions from human activities did not increase in the latter time period compared to the earlier time period for any of the three regions studied here (Fig. 5 and Supplementary Fig. 7). Therefore, local or transported anthropogenic emissions are not driving the observed seasonal pattern changes in CO. The PNW fire CO emissions are the only source that displays the same seasonal pattern change as the observations. Global modeling between 2002 and 2018 further supports the impact of PNW wildfires by showing simulated increases in August CO abundance over North America after 2011 are due to fire emissions from the PNW (Supplementary Information Section 1). While we find that emissions are changing in the PNW and impacting downwind regions, quantifying the role of wildfires on atmospheric composition is complex. For instance, year-to-year variability in transport to the downwind regions may also be contributing to the observed atmospheric variability. The role of emission trends versus emission variability driven by local climate and weather processes such as drought and lightning, as well as the relative contribution of emission increases compared to dynamic changes is left for

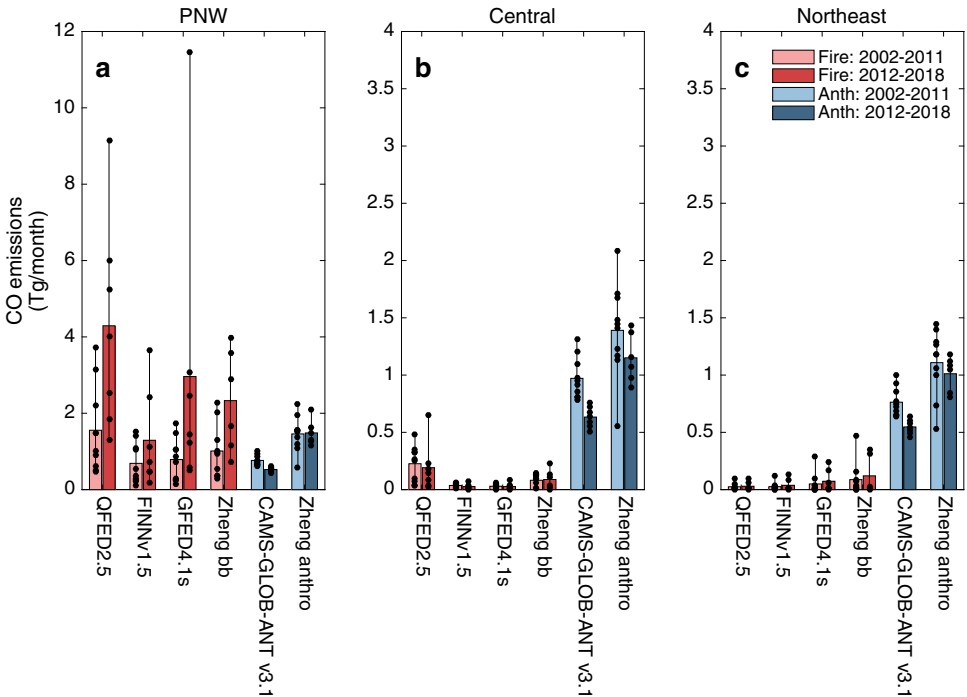

**Fig. 5 Regional comparison of August carbon monoxide (CO) emissions.** Mean August CO emissions are shown in terragrams (Tg) for the two time periods (2002–2011, light colors, and 2012–2018, dark colors) for the three study regions: **a** PNW, **b** Central USA and **c** Northeast. We assess wildfire emissions from four inventories (Fire, red): QFED2.5, FINN1.5, GFED4.1s, and Zheng reanalysis, and two anthropogenic inventories (Anth, blue): CAMS-GLOB-ANT v3.1 and Zheng reanalysis. Note, Zheng reanalysis ends in 2017. Black circles represent regional August means for each year within the time periods.

future work. We conclude that emissions from increasing PNW wildfires are significantly altering the seasonal cycle of atmospheric CO over North America.

## Discussion

Increases in the MOPITT August surface layer CO shows that the PNW wildfires have the potential to impact surface air quality, even at large distances downwind of the wildfires (Fig. 3). Other wildfire-emitted and photochemically produced species, such as the toxic and highly reactive hydrocarbons furan, benzene, and formaldehyde, travel in the pollution plume with CO[39]. Therefore, we expect the health-relevant species surface ozone and fine particulate matter (2.5 μm diameter or smaller, $PM_{2.5}$) to be influenced by the seasonal changes induced by PNW wildfires. Global modeling using the Community Atmosphere Model with chemistry (CAM-chem) during August of 2018 confirms that PNW fire emissions in 2018 impacts surface CO concentrations over large regions of North America (Supplementary Information Section 1). Ozone and $PM_{2.5}$ also show broad modeled surface impacts from the 2018 PNW fires (Supplementary Information Section 1).

These results suggest that similar future years with high PNW wildfire emissions could impact surface air quality over large regions of North America. This is concerning because around 130 million people or more could be detrimentally impacted by wildfire pollution from the PNW, comprised of ~34 million in the PNW, ~23 million in Central USA and ~72 million in the Northeast[28]. Additionally, as the timing and location of the wildfire peak may change in coming years, for example, to include larger emissions from the California region in later months[40], the potential for other months and more people to be impacted by bad air quality increases. Characterizing and

preparing for air quality degradation from wildfire emissions is imperative for minimizing future health impacts.

Air quality degradation due to wildfires has well-established consequences for human health through exposure to fine particulate matter and ozone[17]. Wildfire smoke increases health risks related to acute respiratory conditions[41,42] and evidence is emerging for an adverse effect on cardiovascular conditions[42,43], and pregnancy outcomes[44]. Although the mortality increases from short-term exposure to ambient $PM_{2.5}$ are well established[45], studies specific to mortality associated with wildfire smoke exposure are limited[46–48]. For individuals with existing co-morbidities (e.g., end-stage kidney disease and asthma), wildfire $PM_{2.5}$ has been found to adversely impact mortality[47,48], with limited data of effects for non-accidental mortality downwind of local wildfire events[46,49]. Anthropogenic pollution produces health impacts between states[50] and we assume that the similarly transported wildfire pollution can also have inter-state health impacts. Consequently, we briefly investigate a potential link between transported PNW wildfire pollution and mortality by analyzing seasonality in respiratory deaths for the Central USA state of Colorado for 2002–2011 compared with 2012–2018 (Supplementary Information Section 2). We found evidence that mortality due to chronic respiratory conditions has seen a statistically significant increase in August in the latter period. Although Colorado also experiences local wildfires, the local wildfire season is generally earlier and does not see differences between time periods (Central USA wildfire emissions in Figs. 4 and 5), suggesting instead a relationship to the changing wildfire emissions transported from PNW. While a full epidemiological and attribution study is beyond the scope of this paper, the suggested relationship provides motivation to further study morbidity and mortality across all of North America in relation to increasing PNW wildfire emissions. Globally, concentration

response functions for PM$_{2.5}$ have suggested mortality from wildfire pollution to be 260,000–600,000 annually[51]. A recent health impact assessment using similar concentration response functions found that mortality related to increasing wildfire pollution is predicted to double by 2100 compared to the beginning of the century[52]. Our preliminary study suggests that we may already be seeing these consequences.

While anthropogenic pollution has been decreasing over the last two decades, wildfires in the PNW are an increasingly important source of pollution affecting North American air quality. We found that CO seasonal cycles across large regions of North America have been significantly altered due to an increase in PNW wildfires, and observe increases in atmospheric pollution in August. Wildfire pollution can impact air quality at the surface both locally and downwind, with potential detriments to human health[41,43,53]. Since amplified PNW wildfires are predicted under accelerating climate change[5,7] it is essential to understand both local and transported impacts on air quality in North America.

## Methods

**Satellite-measured CO and AOD**. We use measurements from the NASA/Terra satellite, launched in December 1999. Terra completes a sun-synchronous orbit, crossing the equator at ~10:30 am and pm local time[54].

MOPITT CO: Measurements Of Pollution In The Troposphere (MOPITT) is a nadir-viewing gas correlation radiometer measuring in the thermal infrared (TIR) near 2140 and at 4275 cm$^{-1}$ in the near-infrared (NIR)[24]. Global coverage occurs about every three days with a ground resolution of ~22 km$^2$ at nadir. Optimal estimation on gas cell absorption retrieves CO profiles of dry air VMR on 10 vertical layers, which are integrated to column amounts[55]. We use version 8, joint TIR and NIR retrievals (Level 2 and Level 3) (https://doi.org/10.5067/TERRA/MOPITT/MOP02J_L2.008), that includes dataset improvements. MOPITT CO has been validated against NOAA aircraft and long-term ground-based FTIR stations to find no substantial drift in the measurements[55–57]. In V8, there is a negligible drift of −0.001 ± 0.070% per year for CO total column amounts and 0.02 ± 0.08% per year for surface layers[55]. We use daytime retrievals over land scenes, filtered to stringent anomaly diagnostics, signal-to-noise in the 5A channel must be greater than 1000 and in 6A greater than 400, and pixel 3 is removed because of the large noise variability[58]. We find these filtering criteria are important for maximizing the significance of our findings. Data from 2002 onwards is used to avoid discontinuities from the MOPITT optical board failure that occurred in 2001[59].

MODIS AOD: The Moderate Resolution Imaging Spectroradiometer (MODIS) is a passive imaging radiometer, measuring reflected solar and thermal radiation in 36 bands, with global coverage in ~1 day at spatial resolution between 250 m and 1 km at nadir. We use AOD at 550 nm from the merged Dark Target Dark Blue (DTDB) product[27]. Retrievals must pass recommended quality assurance[60]. Monthly global products from Collection 6.1 (C6.1) Level-3 MODIS (MOD08_M3) are used (https://doi.org/10.5067/MODIS/MOD08_M3.061), at 1° × 1° spatial resolution[61]. C6.1 is largely free of artificial drifts due to updates in calibration stability that mitigate an observed drift in radiance and reflectance[23,62].

**Emission inventories**. Fire emitted CO is investigated using four emission inventories calculated with varying methods. Although these different fire emission inventories all use the same satellite instrument, MODIS, to detect fires, they diverge on how that information is used, providing a range of estimates that captures a range of uncertainties[63]. We use these different inventories in our study to account for a wide range of uncertainties, lending confidence to our assertion that changes in CO are driven by fire emissions. The Fire INventory from NCAR version 1.5 (FINN1.5) is based on MODIS satellite observations of active fires with corresponding pre-assumed burned area[34]. Global Fire Emissions Database version 4.1 with small fires (GFED4.1s) uses MODIS satellite observations of burned area[35] with adjustments for small fires[64]. Both FINN1.5 and GFED4.1s multiply burned area with biomass loading, combustion completeness, and emission factors to create emissions. In contrast, the Quick Fire Emissions Dataset version 2.5 (QFED2.5) uses MODIS fire radiative power (FRP) multiplied by biome-specific scaling and emission factors[36,65]. Finally, we use emissions based on multi-species atmospheric inversions of satellite retrieved composition (e.g., MOPITT CO), that uses GFED 4.1s as prior information[37] that we name Zheng reanalysis. Anthropogenic emissions of CO are also investigated with the Zheng reanalysis[37], as well as with a second inventory, the Copernicus Atmosphere Monitoring Service Global Anthropogenic version 3.1 (CAMS-GLOB-ANT v3.1)[38].

**Regions of interest**. Regions are defined as Pacific Northwest (PNW, 38°–57°N, 127°–110°W), Central USA (35°–49°N, 110°–95°W), and the Northeast (41°–53°N, 95°–74°W), outlined in Fig. 1. The PNW covers the majority of wildfires that

occurred in the region, and encompasses the upward August trend in CO. We aimed to contain the main transport areas shown in Fig. 1 in Central USA and Northeast region boundaries.

**Analysis methodology**. The spatial trend plots between 2002 and 2018 are created by performing ordinary least squares analysis on the MOPITT Level 3 gridded 1° × 1° monthly products for CO, and level 3 MODIS monthly global products for AOD. Significant non-zero trends are determined using two-tailed t-tests, α = 0.05.

The regional time series analysis uses MOPITT Level 2 retrievals, which are converted into column average VMR (X$_{CO}$) by dividing retrieved column CO by the respective reported dry air column. All X$_{CO}$ and surface CO retrievals are collected within a region for a particular month and the monthly mean, standard deviation, 25th, median and 75th percentiles are calculated.

Monthly regional time series of average CO are detrended between 2002 and 2018 using the Northern Hemisphere background trend of −0.57% (−0.50 ppb) per year[23]. AOD is not detrended because the large counteracting regional trends and small influence from long-range transport do not lend themselves to meaningful global trends.

CO and AOD time series are split into two time periods: 2002–2011 and 2012–2018 and average seasonal cycles and standard deviation are determined. This split is based on observing consistent changing behavior between time periods for the three regions (Supplementary Fig. 5). Uncertainty ranges are plus or minus one standard deviation of the monthly estimates. Independent t-tests (two-tailed: α = 0.05) were carried out to determine significant differences between the monthly averages in the seasonal cycles between the two time periods. Effect sizes of significant findings were calculated using Cohen's d. Summary statistics related to the observed August peak differences in Fig. 3a–i, are presented in Table 1.

**Reporting summary**. Further information on research design is available in the Nature Research Reporting Summary linked to this article.

## Data availability

Data used in this analysis are publicly available. The raw MOPITT CO data are available from NASA under the accession code (https://doi.org/10.5067/TERRA/MOPITT/MOP02J_L2.008). The raw MODIS AOD data are available from NASA under the accession code (https://doi.org/10.5067/MODIS/MOD08_M3.061). The raw emission inventory CO data are available from their respective repositories: QFED2.5 – https://portal.nccs.nasa.gov/datashare/iesa/aerosol/emissions/QFED/v2.5r1/0.25/QFED/; FINN1.5 – http://bai.acom.ucar.edu/Data/fire/; GFED4.1s – https://globalfiredata.org/pages/data/; CAMS-GLOB-ANT v3.1 – accessed here from https://eccad3.sedoo.fr/, archive available from https://ads.atmosphere.copernicus.eu/cdsapp#!/dataset/cams-global-emission-inventories; Zheng Reanalysis – https://doi.org/10.6084/m9.figshare.c.4454453.v1. The CAM-chem model output used to analyze 2002–2018 is available from the NCAR Research Data Archive, with accession code 10.5065/CKR4-GP38 (https://rda.ucar.edu/datasets/ds313.7). The MODIS burned area and fire count Climate Modeling Grid data products are available from the University of Maryland via sftp from fuoco.geog.umd.edu. The data generated in this study for Figs. 2–5 are provided in the Source Data file. The raw mortality data can be obtained from the via Wide-ranging OnLine Data for Epidemiologic Research (WONDER) from the National Vital Statistics System at the National Center for Health Statistics, Centers for Disease Control, available at https://wonder.cdc.gov/Deaths-by-Underlying-Cause.html. Source data are provided with this paper.

## Code availability

Code used to analyze MOPITT and MODIS trends, as well as seasonal cycles is publicly available via GitHub[66].

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

## Acknowledgements

This material is based upon work supported by the National Center for Atmospheric Research (NCAR), which is a major facility sponsored by the National Science Foundation under Cooperative Agreement No. 1852977. Any opinions, findings, and conclusions or recommendations expressed in this publication are those of the author(s) and do not necessarily reflect the views of the National Science Foundation. The NCAR MOPITT project is supported by the National Aeronautics and Space Administration (NASA) Earth Observing System (EOS) Program. The MOPITT team also acknowledges support from the Canadian Space Agency (CSA), the Natural Sciences and Engineering Research Council (NSERC) and Environment Canada, and the contributions of COM-DEV (the prime contractor) and ABB BOMEM. MODIS is a sensor aboard NASA's Terra satellite mission. Native resolution L1B, L2 aerosol retrievals, and the gridded L3 atmospheric product used here (MOD08_M3) are all distributed via the LAADS DAAC (https:ladsweb.modaps.eosdis.nasa.gov). We would like to acknowledge high-performance computing support from Cheyenne (doi:10.5065D6RX99HX) provided by NCAR's Computational and Information Systems Laboratory, sponsored by the National Science Foundation. We thank the Colorado Department of Public Health & Environment for providing mortality data. Figures 1–5 were improved after consultation with Simmi Sinha. We thank Judith Buchholz for statistics consultation. We also thank NCAR internal reviewers Siyuan Wang and Rebecca Hornbrook.

## Author contributions

R.R.B. and H.M.W. developed the work; R.R.B., M.P., and W.T. performed the analysis; T.S. performed CAM-chem simulations, B.G. evaluated model results, M.R. performed health relationship analysis, S.M. provided input on health analysis interpretation, B.Z. created and advised on the use of reanalysis emissions, R.R.B interpreted results and drafted the manuscript; H.M.W., D.P.E., and M.N.D. helped interpret results, edited the manuscript, and provided input for the Discussion and Conclusion section, M.C. and R.C.L. helped interpret results, and advised on the use of MODIS AOD.

## Competing interests

The authors declare no competing interests.
