## [Peer Review File · Nature Communications]

New seasonal pattern of pollution emerges from changing North American wildfiresREVIEWER COMMENTS

Reviewer #1 (Remarks to the Author):

The authors find a shift in the seasonal CO pattern and trend resulting from wildfires and attribute it to wildfires from the PNW. This the change in AOD that was previously published. The shift is quite striking the in the month of August and is linked to the changes in wildfire emissions driven by climate change. The authors further explore the connection to air quality and health downwind.

This is a well-written and convincing paper. I feel it is suitable in scope and impact for Nature-Communications.

Four FF emission databases are used to show the predicted different in emissions for the two periods considered, and they all agree, more or less, on the relative change. My only qualm is the contention that these emission inventories are independent (“Independently created fire and anthropogenic emission inventories are used to support that PNW fire is driving the observed seasonal pattern changes.”) They are (to my knowledge) all fundamentally driven by MODIS (or the like) thermal anomaly hotspot data. While some may also use FRP, and some other quantities, the shift in fire emissions from the four databases between the two periods shown in Figure 5 are all fundamentally driven by the hot spots (ie, timing of fires, location of fires, some indicator of intensity), and thus not really independent. The differences lie more in the details such as emission factors, rate of spread, etc...). I would say this does not appreciably weaken the link appreciably. This needs to be qualified/clarified.

If this one point is addressed then I recommend immediate publication.

Reviewer #2 (Remarks to the Author):

This is a very well written manuscript addressing the timely issue of wildfires and their impact on air quality and human health. The analysis presented is thorough and clearly explained in a clear and understandable manner. However, there are a couple of points that I think will be good to address before accepting for publication.

The two main points to address, which I think will strengthen the analysis and be of benefit to the reader are to add:

1. A brief analysis of the trend in fire activity based on observed fire counts across the region. This could help to further isolate the role of fires in the observed August peaks of column average CO and AOD and I recommend including a time series of fire counts, or similar metric, to compliment Figure 2. This could also be added in the supplement if needed. Also, if the authors can comment on how much the increasing CO trend is influenced by an increase in the frequency of enhanced August fire activity as it seems from Figure 2 that the August peak is considerably higher in a handful of the last decade or so but not in all years.

2. Some additional discussion on the role of the synoptic meteorological conditions, the influence they have on both the fire activity and pollution transport, and whether they are prevalent conditions or change during those years with the highest August activity. Some comments on this, especially in relationship to other states which experience significant fire activity, such as in California which is mentioned on page 9, will be very informative and help to highlight the complexity in understanding regional trends in fire and pollution, and the impact on health downwind of the source.

I don't believe that any new significant new analysis is required but some acknowledgement of these points will be helpful.

Specific comments are below:

Line 36: clarify that CO is a good tracer for tracking atmospheric pollution in general – I agree that it is valuable for tracking fire pollution but it currently reads as though it is not the case for other pollution sources.

Line 41: suggest rephrasing to “a recent slow-down in the decreasing trend of Northern Hemisphere CO”.

Line 45: suggest changing “local and distant” to “near- and far-field”, which is sometimes used in this context.

Line 61: it may be useful to add something brief about background CO levels, and the role of long-range transport pathways, although I think the information presented in the manuscript and supplement provide strong evidence for the main driver being increased fire activity. I believe this may be more the case in spring but do the authors think LRT can be completely discounted in August?

Line 80 & Figure 2: picking out the peaks is a bit of a challenge with the size of the Figure – if some faint vertical lines could be added to mark out the start of each year then I think it will be a lot easier to interpret.

Line 119, and throughout: please check the consistent use of “fires” and “wildfires”.

Line 136: the Copernicus Atmosphere Monitoring Service should refer to the whole service, and not just the emissions inventories that have been developed. The emissions inventory should be named as Copernicus Atmosphere Monitoring Service Global Anthropogenic (CAMS-GLOB-ANT) emissions. Line 150: it will be useful, and informative to the reader, to add a brief statement on the range of air pollutants – of course the most prevalent pollutants (ozone, PM_{2.5}) will be increased but wildfire smoke includes many other toxic air pollutants which may not always be so prevalent in some population centres.

Line 161-162: I wonder if it is the case that there is really a clear shift in the persistence of August fire activity in the Pacific Northwest or if it is more the case that the data is showing an increase in the frequency of August fires in recent years (as not all years are showing such a pronounced August peak) and it will be informative if more analysis could be presented in relation to this. For California is it the case that there is a lot more variability in the timing of wildfire activity throughout the year which makes a persistent impact on air quality more challenging to establish? Some comments on this, and especially on differences in the meteorology affecting both PNW fires and air quality impacts, and differences relative to the same in California. The influence of meteorology on air quality impacts can be critical and I think the current manuscript could be improved by including some aspects of this in the analysis.

Line 223: anthropogenic emissions inventories seem to be missing from this section?

Line 260: please include the version number of the CAMS-GLOB-ANT emissions inventory – I understand that the authors will have accessed the data from the ECCAD database but versions kept there may change and the official database will be moved to the CAMS Atmosphere Data Store (<https://ads.atmosphere.copernicus.eu/cdsapp#!/dataset/cams-global-emission-inventories?tab=overview>).

Figure 2: please add faint vertical lines to mark out each year in the time series to make it easier to interpret. I also recommend adding a time series of fire counts to highlight the increased frequency of August activity to correspond with the peaks in CO.

Reviewed by Mark Parrington (ECMWF)

Reviewer #3 (Remarks to the Author):

General Comments:

The authors have shown increasing wildfires and associated CO emissions in the Pacific Northwest that propagate from west to east with detectable peaks in August in the central and a shift in the peak in the northeastern US. The authors posit that the increased CO concentration down-wind from the origin of the fire along with contemporaneous increases in satellite-derived aerosol optical density suggest that impaired air quality from remote wildfires in the Pacific Northwest could be affecting the health of people in the Central and Northeastern US. The study is further enhanced by associating respiratory mortality to the increase in wildfire emissions in Colorado for 2002-2011 compared to 2012-2018.

Specific Comments:

Pg. 2. L. 20-22. Apart from the climate related effects driving increases in the extent and expansion of the wildfire season, another factor that is often overlooked is 100 years of land management policy in the U.S. that prioritized suppression of fire and limited prescribed fire. These policies that have resulted in forest overgrowth, susceptibility to disease and insect infestation have resulted in abundant fuel for wildfire and a lack of natural firebreaks. The land management policy contribution to the current wildfire crisis is not the focus of the paper, but to the extent you call for actions to reduce wildfire and protect health those actions will need to include a cohesive strategy for land management.

Pg. 2. L. 32-33. Consider modifying the following sentence "The influence of PNW fires on other atmospheric trace gas pollutants, as well as the down-wind impacts on air quality, requires further investigation." to "The influence of PNW fires on other atmospheric trace gas and aerosol pollutants, as well as the down-wind impacts on air quality and human health, requires further investigation." In your study CO essentially is serving as a surrogate indicator of other emission pollutants, primarily smoke (PM) that can have human health effects. Introducing aerosols in the previous sentence will strengthen the role of CO as stated in L. 34-36. It's important to get this point made early in the paper as one would not anticipate any health effect of CO at the levels measured. The EPA has established two primary National Ambient Air Quality Standards for carbon monoxide (CO), one averaged over eight-hours (9 ppm not to be exceeded more than once per year) and a one-hour averaging time (35 ppm not to be exceeded more than once per hour). As shown in Figure 2 the peak concentrations of CO from 2002 to 2018 are approximately 120 ppb, a value that is about 75 times less than the standard for the 8-hour average.

Pg. 5. L. 86-87. What was the reasoning to separate the time series into the two intervals chosen? Based on Pg. 7. L. 132 was the choice based on an empiric observation that the "Inventories also show enhanced August fire CO in the PNW for CO in the PNW for 2012-2018 to 2002-2011?"

Pg. 6. L. 96-106. The presentation of seasonal cycles of atmospheric CO concentration is clear and well done.

Pg. 7. L. 130. "pf" should read "of".

Pg. 9. L. 166-168. When describing the health risks associated with wildfire smoke exposure your statements express a high level of certainty in the associations. While this unequivocally true for respiratory disease, the associations with cardiovascular and birth outcomes are as of now not fully defined. Recommend softening the statements on health effects to say something like, "Wildfire smoke increases health risks related to acute respiratory³⁹, ⁴⁰ and evidence is now emerging suggesting an adverse effect on cardiovascular conditions⁴⁰, ⁴¹, and pregnancy outcomes⁴²."

Pg. 9. L. 168-169. With respect to the following statement "Although the mortality increases from short-term exposure to ambient PM_{2.5} are well established⁴³, studies specific to short-term wildfire smoke exposure to smoke are limited." consider the following literature:

- 1: Doubleday A, et al. Mortality associated with wildfire smoke exposure in Washington state, 2006-2017: a case-crossover study. *Environ Health*. 2020 Jan 13;19(1):4. Doi: 10.1186/s12940-020-0559-2.

- 2: Matz CJ, et al. Health impact analysis of PM2.5 from wildfire smoke in Canada (2013-2015, 2017-2018). *Sci Total Environ*. 2020 Jul 10;725:138506. doi: 10.1016/j.scitotenv.2020.138506.
- 3: Xi Y, Kshirsagar AV, Wade TJ, Richardson DB, Brookhart MA, Wyatt L, Rappold AG. Mortality in US Hemodialysis Patients Following Exposure to Wildfire Smoke. *J Am Soc Nephrol*. 2020 Aug;31(8):1824-1835. doi: 10.1681/ASN.2019101066.
- 4: Liu Y, Austin E, Xiang J, Gould T, Larson T, Seto E. Health Impact Assessment of PM2.5 attributable mortality from the September 2020 Washington State Wildfire Smoke Episode. *medRxiv [Preprint]*. 2020 Sep 22:2020.09.19.20197921. doi: 10.1101/2020.09.19.20197921.
- 5: Casey JA, Kioumourtzoglou MA, et al. Wildfire particulate matter in Shasta County, California and respiratory and circulatory disease-related emergency department visits and mortality, 2013-2018. *Environ Epidemiol*. 2020 Dec 21;5(1):e124. doi:10.1097/EE9.0000000000000124.
- 6: Haikerwal A, et al. Impact of Fine Particulate Matter (PM2.5) Exposure During Wildfires on Cardiovascular Health Outcomes. *J Am Heart Assoc*. 2015 Jul 15;4(7):e001653. doi: 10.1161/JAHA.114.001653. PMID: 26178402; PMCID: 10.1161/JAHA.119.014125.
- 7: Jones CG, et al. Out-of-Hospital Cardiac Arrests and Wildfire-Related Particulate Matter During 2015-2017 California Wildfires. *J Am Heart Assoc*. 2020 Apr 21;9(8):e014125. doi: 10.1161/JAHA.119.014125.

Pg. 9. L. 171. The following clause is difficult to understand "... has so far not revealed an effect for general populations downwind of local wildfire events⁴⁵." given that it is among the general population that at-risk individual live and are effected.

We thank all the reviewers for their time spent in reading this manuscript and for their thoughtful and valuable comments and suggestions. Below we address Reviewer comments noted in blue with our responses shown in black. Responses refer to line numbers and reference numbers in the updated manuscript.

Reviewer #1 (Remarks to the Author):

The authors find a shift in the seasonal CO pattern and trend resulting from wildfires and attribute it to wildfires from the PNW. This the change in AOD that was previously published. The shift is quite striking the in the month of August and is linked to the changes in wildfire emissions driven by climate change. The authors further explore the connection to air quality and health downwind.

This is a well-written and convincing paper. I feel it is suitable in scope and impact for Nature-Communications.

Thank you for the general assessment and feedback.

Four FF emission databases are used to show the predicted different in emissions for the two periods considered, and they all agree, more or less, on the relative change. My only qualm is the contention that these emission inventories are independent ("Independently created fire and anthropogenic emission inventories are used to support that PNW fire is driving the observed seasonal pattern changes.") They are (to my knowledge) all fundamentally driven by MODIS (or the like) thermal anomaly hotspot data. While some may also use FRP, and some other quantities, the shift in fire emissions from the four databases between the two periods shown in Figure 5 are all fundamentally driven by the hot spots (ie, timing of fires, location of fires, some indicator of intensity), and thus not really independent. The differences lie more in the details such as emission factors, rate of spread, etc...). I would say this does not appreciably weaken the link appreciably. This needs to be qualified/clarified.

If this one point is addressed then I recommend immediate publication.

Thank you for identifying this potential miscommunication of using "independently created". We have altered the text to use "different inventories".

L49: "Different fire and anthropogenic emission inventories are used to support..."

L136-137: "is supported by four different global fire emission inventories:..."

Although these different inventories all use the same instrument (MODIS) for detecting fires, they diverge on how that information is used. The inventories use different data products from MODIS (e.g., Fire count, Burned Area, Fire Radiative Power) as well as different assumptions about emissions factors, aggregated vegetation types, and estimation of fuel burnt in the creation of trace gas and aerosol emissions. Specifically, FINN1.5 uses fire count, GFED4.1s uses burned area adjusted with fire counts for small fires, and QFED2.5 uses FRP (Table R1). Each inventory also uses different land cover datasets and different aggregation of biomes. Additionally, the reanalysis product (Zheng et al., 2019), assimilates MOPITT CO and WDCGG in situ Methyl Chloroform to estimate emissions from anthropogenic and biomass burning source sectors, thereby correcting the prior fire emissions from GFED4.1s.

The result is that we are intrinsically comparing fire count, fire radiative power and burned area, with added uncertainties. It is striking that we can still identify a change on top of the uncertainty afforded by different methodologies. We now discuss this uncertainty in the manuscript on L242-246:

"Although these different fire emission inventories all use the same satellite instrument, MODIS, to detect fires, they diverge on how that information is used, providing a range of estimates that captures a range of uncertainties (63) . We use these different inventories in our study to account for a wide range of uncertainties, lending confidence to our assertion that changes in CO are driven by fire emissions."

Table R1: Different sources of products used by different inventory algorithms to create fire emissions, that leads to a range of estimates that covers a range of uncertainties.

	Inventory		
	FINN1.5 (34)	GFED4s (34,64)	QFED2.5 (36,65)
Fire detection	Fire Count: MOD14 with 1km ² assumed burned except in Savanna /Grasslands where 0.75 km ² assumed burned	Burned area and fire count: MCD64A1, MOD14A1 Collection 5 and Collection 6	Fire radiative power: MOD14 (with MOD03 Geolocation)
Fuel load	Hoelzemann et al. (2004) with updates.	Carnegie–Ames–Stanford Approach (CASA) model for NPP and carbon pool estimation. Adjusted turnover rates.	FRP does not require these parameters. Instead uses biome dependent scaling factors to relate FRP to dry mass consumed.
Combustion completion (CC)	Relative to tree cover fraction Ito and Penner (2004) $\geq 60\%$ tree cover, woody _{CC} = 0.3; herbaceous _{CC} = 0.9 $< 40\%$ tree cover, woody _{CC} = 0; herbaceous _{CC} = 0.98 40–60 % tree cover, woody _{CC} = 0.3; herbaceous _{CC} = $e^{-0.13 \text{frac.trees}}$	Uses a min/max range of CC values for different vegetation types (van der Werf et al., 2010), scaled by soil moisture. Includes adjustments for tree mortality.	
Vegetation/ Land cover	MODIS LCT with IGBP classification and MODIS VCF. Bare cover further scales burned area.	MOD44B VCF for fractional tree cover in 2013 adjusted for earlier years by deforestation rates. Annual LCT from MCD12C1 with Friedl et al., (2010) classification.	IGBP-INPE
Biome aggregation	7 types: Savanna /Grasslands; Woody Savannas and Shrublands; Tropical Forest; Temperate Forest; Boreal Forest; Cropland; Peat	7 types: Savanna, grassland, & shrubland; Boreal forest; Temperate forest; Deforestation and degradation; Peatland; Agricultural waste burning	4 types: Tropical Forest; Extratropical Forest; Savanna/Shrublands ; Grasslands
Emission	Akagi et al. (2011) with	Akagi et al. (2011)	Andreae and Merlet

Factors	updates		(2001)
Clouds and coverage	Tropical Region: Fire continues the next day at 50% size	Combine burned area with fire persistence (calculated using active fires) to better estimate tropical deforestation fires.	Cloud-clearing sequential approach: damped persistence

Reviewer #2 (Remarks to the Author):

This is a very well written manuscript addressing the timely issue of wildfires and their impact on air quality and human health. The analysis presented is thorough and clearly explained in a clear and understandable manner. However, there are a couple of points that I think will be good to address before accepting for publication.

We appreciate your overall assessment and detailed feedback. We address specific comments below.

The two main points to address, which I think will strengthen the analysis and be of benefit to the reader are to add:

1. A brief analysis of the trend in fire activity based on observed fire counts across the region. This could help to further isolate the role of fires in the observed August peaks of column average CO and AOD and I recommend including a time series of fire counts, or similar metric, to compliment Figure 2. This could also be added in the supplement if needed.

- Our study intrinsically compares fire count, fire radiative power and burned area, by using the different inventories based on each of these products. However, as suggested to further connect the August peaks in CO with fire variability, we now also include an image of month average PNW CO against burned area and fire count time series as an Extended Data Figure 4 (Fig. R1 below).

Figure R1 (& Extended Data Fig. 4): Time series of PNW monthly average MOPITT CO (blue) and monthly summed MODIS fire detection for burned area (red, MCD64CMQ(66)) and fire

count (black, MOD14CMQ(67)) in the same region. Vertical lines show the beginning of each year.

We have added to following to the manuscript on L89:

"These secondary CO peaks also coincide with peak burning in the PNW, as described by MODIS fire count and burned area (Extended Data Figure 4), further supporting a link between wildfires and the CO August peak. The magnitude of peak PNW burning is generally larger in 2012-2018 compared to 2002-2011."

Also, if the authors can comment on how much the increasing CO trend is influenced by an increase in the frequency of enhanced August fire activity as it seems from Figure 2 that the August peak is considerably higher in a handful of the last decade or so but not in all years.

We agree that there is large interannual variability in the August CO peaks during the recent decade, likely driven by climate variability through impacts on wildfire. We performed the Student's T-Test to give a measure of the significance of changes in CO despite this interannual variability, and found August changes are significant. Note that prior to 2011, most of the years show a single peak in the spring (except 2006 and 2010), while after 2011, a substantial secondary peak in August occurs every year except 2016. We also tested 2017, 2016 as end points in the August trend analysis and still found increasing trends (Figure R2). For these reasons we believe the August feature is driven by a general trend upward.

Figure R2: August trend in month average satellite-measured CO from 2002-2016 (left) and 2002-2017 (right).

Quantifying fire emissions changes on the resulting CO abundance change would require a modeling sensitivity study, which we leave for future work.

2. Some additional discussion on the role of the synoptic meteorological conditions, the influence they have on both the fire activity and pollution transport, and whether they are prevalent conditions or change during those years with the highest August activity. Some comments on this, especially in relationship to other states which experience significant fire activity, such as in California which is mentioned on page 9, will be very informative and help to highlight the complexity in understanding regional trends in fire and pollution, and the impact on health downwind of the source.

Thank you for highlighting these two issues:

(i) Local meteorological conditions, such as temperature, relative humidity and lightning, can drive wildfire activity variability in the PNW. Both climate variability and climate trends can impact meteorological conditions (e.g., drought) that feedback into the fire conditions. For example, the El Niño Southern Oscillation impacts interannual variability in precipitation in North America. The interannual variability is added on top of the long-term trend. Research has found that a warming climate has increased probability of fires in the PNW region through impacting drought and we mention several references relating to this trend in the introduction. Attributing the wildfire variability to weather variability is beyond the scope of this work.

(ii) Changes in long-range transport have the potential to influence atmospheric pollution amounts transported downwind from the PNW. Over North America, this transport is largely determined by the Jet Stream. Seeing as we observe significant changes in the August average CO between two time periods, we analyse corresponding average wind patterns. Figure R3 aggregates MERRA2 reanalysis winds for August 2002-2011 compared to 2012-2018 and suggests no major changes in transport between time periods studied here. Although small changes in long-range transport could add some interannual variability to CO, the large aggregate regions used in our study would minimize the impact. To quantify the importance of emissions changes relative to transport changes it would be valuable to complete a model sensitivity study where emissions are held constant, and meteorology is allowed to change. We leave this for future work.

Figure R3: August composite MERRA2 wind fields at approximately 500hPa (top row) and approximately 800 hPa (bottom row) for 2002-2011 (first column) and 2012-2018 (second column).

I don't believe that any new significant new analysis is required but some acknowledgement of these points will be helpful.

We have added the following to our manuscript lines L154:

"While we find that that emissions are changing in the PNW and impacting downwind regions, quantifying the role of wildfires on atmospheric composition is complex. For instance, year-to-year variability in transport to the downwind regions may also be contributing to the observed atmospheric variability. The role of emission trends versus emission variability driven by local climate and weather processes such as drought and lightning, as well as the relative contribution of emission increases compared to dynamic changes is left for future work."

Specific comments are below:

Line 36: clarify that CO is a good tracer for tracking atmospheric pollution in general – I agree that it is valuable for tracking fire pollution but it currently reads as though it is not the case for other pollution sources.

- We have changed Line 33 to:

"With an atmospheric lifetime ranging from weeks to months, CO is valuable for tracking the atmospheric transport of large sources of pollution, such as from wildfires."

Line 41: suggest rephrasing to "a recent slow-down in the decreasing trend of Northern Hemisphere CO".

- Rephrased as suggested (L39).

Line 45: suggest changing "local and distant" to "near- and far-field", which is sometimes used in this context.

- Changed as suggested (L44).

Line 61: it may be useful to add something brief about background CO levels, and the role of long-range transport pathways, although I think the information presented in the manuscript and supplement provide strong evidence for the main driver being increased fire activity. I believe this may be more the case in spring but do the authors think LRT can be completely discounted in August?

- Figure 1(a) and Extended Data Figure 2 shows an August decreasing trend in column CO from the west, over the Pacific Ocean, which is evidence of the global average decrease in CO. Furthermore, the positive trend in August CO is strongest over the PNW, globally. In our Supplementary Information Section 1 we performed a model study that showed no increases over the inflow region to North America (Supp. Fig. 1.4). Consequently, from this combined evidence we postulate that external transport pathways do not play a large role in driving the seasonal change in North American CO. We have expanded our discussion around the potential transported impacts L62:

"To the west of the PNW over the Pacific Ocean, negative trends in CO are observed in August. Recent work(23) identified downward trends in the Northern Hemisphere background CO in all months as well as strong downward trends in CO over Northeast China,

suggesting transported pollution into North America via westerly flow does not play a large role in the August positive CO trend."

Line 80 & Figure 2: picking out the peaks is a bit of a challenge with the size of the Figure – if some faint vertical lines could be added to mark out the start of each year then I think it will be a lot easier to interpret.

- We have altered the Figure 2 and Extended Data Figure 4 to more clearly show each year with vertical grey lines.

Line 119, and throughout: please check the consistent use of "fires" and "wildfires".

- We have checked the use of "fire/s" versus "wildfire/s" and changed the usage of fire to wildfire in most locations, as this is the main type of fire discussed in the study (as opposed to deforestation or agricultural fires). However, several exceptions occur: when describing that CO sources include fire (Line 32) because CO is sourced from any type of fire, a global decline in tropical fires (Line 39) as many tropical fires are deforestation fires; when in the title of the emission inventories (for example The Fire INventory from NCAR); where we discuss the MODIS "active fires" or "fire radiative power" products in the Methodology section (Line 247 and 252); where we describe that "small fires" are added in the GFED global product (Line 249) because these may include deforestation and agricultural fires rather than just wildfires; when discussing the "fire emissions" specifically from inventories or in modeling studies, particularly in the Supplementary Section, because these inventories include agricultural or deforestation fire emissions (e.g., Line 49, 139 and Line 512).

Line 136: the Copernicus Atmosphere Monitoring Service should refer to the whole service, and not just the emissions inventories that have been developed. The emissions inventory should be named as Copernicus Atmosphere Monitoring Service Global Anthropogenic (CAMS-GLOB-ANT) emissions.

- We have changed the text accordingly, as well as updated the labels in Figure 5 and Extended Data Figure 7. Lines 145-147 have been changed to:

"Additionally, two anthropogenic emission inventories (Copernicus Atmosphere Monitoring Service Global Anthropogenic emissions, CAMS-GLOB-ANT(38), and Zheng reanalysis(37))..."

Line 150: it will be useful, and informative to the reader, to add a brief statement on the range of air pollutants – of course the most prevalent pollutants (ozone, PM2.5) will be increased but wildfire smoke includes many other toxic air pollutants which may not always be so prevalent in some population centres.

- We altered the sentence Line 164:

"Other wildfire-emitted and photochemically produced species, such as the toxic and highly reactive hydrocarbons furan, benzene, formaldehyde, travel in the pollution plume with CO(38). Therefore, we expect the health-relevant species surface ozone and fine particulate matter (2.5 micron diameter or smaller, PM2.5) to be influenced by the seasonal changes induced by PNW wildfires."

Line 161-162: I wonder if it is the case that there is really a clear shift in the persistence of August fire activity in the Pacific Northwest or if it is more the case that the data is showing an increase in the frequency of August fires in recent years (as not all years are showing such a pronounced August

peak) and it will be informative if more analysis could be presented in relation to this. For California is it the case that there is a lot more variability in the timing of wildfire activity throughout the year which makes a persistent impact on air quality more challenging to establish? Some comments on this, and especially on differences in the meteorology affecting both PNW fires and air quality impacts, and differences relative to the same in California. The influence of meteorology on air quality impacts can be critical and I think the current manuscript could be improved by including some aspects of this in the analysis.

The key result from our research is that there is a secondary CO peak emerging due to fires. The OH-driven CO loading maximum at the end of Northern Hemisphere winter/spring is being supplemented by a western states fire peak at the end of summer. This new peak occurs during the lowest CO background values -- the minimum in CO background occurs in August due to a maximum in OH-driven loss.

We agree that year-to-year variability may shift peak wildfires either earlier and/or later in the PNW wildfire season and also change the area within the PNW (or externally in Southern California) that contributes the most to CO loading. Note that our definition of PNW includes some of Northern California, to around San Francisco (~38 N). The fire emission inventories show that PNW wildfire CO emissions are larger for the whole wildfire season July-September (Extended Data Figure 6) in the recent time period. However, we are only currently observing significant atmospheric changes in the August average CO in the PNW. It is likely that more data would be needed for comparisons in other months when CO has higher background levels.

We have slightly clarified the discussion lines 176:

"Additionally, as the timing and location of the wildfire peak may change in coming years, for example to include larger emissions from the California region in later months(40), the potential **for other months** and more people to be impacted by bad air quality increases."

We further discussed meteorology and transport in response to previous Main Point #2 above and our response to the comment above regarding Line 61.

Line 223: anthropogenic emissions inventories seem to be missing from this section?

- Added to Line 255:

" Anthropogenic emissions of CO are also investigated with the Zheng reanalysis (36), as well as with a second inventory, the Copernicus Atmosphere Monitoring Service Global Anthropogenic version 3.1 (CAMSGLOBANT v3.1)(37)."

Line 260: please include the version number of the CAMSGLOBANT emissions inventory – I understand that the authors will have accessed the data from the ECCAD database but versions kept there may change and the official database will be moved to the CAMS Atmosphere Data Store (<https://ads.atmosphere.copernicus.eu/cdsapp#!/dataset/cams-global-emission-inventories?tab=overview>).

- Version 3.1 added to L257, L284 Data Availability changed to:

"CAM5-GLOB-ANT version 3.1 -- accessed from <https://eccad3.sedoo.fr/>, archive available from <https://ads.atmosphere.copernicus.eu/cdsapp#!/dataset/cams-global-emission-inventories>"

Figure 2: please add faint vertical lines to mark out each year in the time series to make it easier to interpret. I also recommend adding a time series of fire counts to highlight the increased frequency of August activity to correspond with the peaks in CO.

- Addressed according to Main Point #1 and comment above regarding Line 80.

Reviewed by Mark Parrington (ECMWF)

Note in reviewing our manuscript in regards to the comments above we have also edited lines 100-112 to improve clarity :

"In all three regions and for both time periods, CO loading shows a photochemically-driven maximum during Northern Hemisphere spring, in April (Fig. 3 a-c). The CO seasonal cycle results from a combination of source and loss mechanisms, with loss dominated by reaction with the photochemically produced hydroxyl (OH) radical(18). Due to seasonal variability in sunlight, the chemical lifetime of CO over winter is about 2 months, compared to less than a month in summer when photochemical production of OH is at a maximum(29). Consequently, in atmospheres with well-mixed atmospheric conditions (i.e. homogeneous properties) that are distant from sources, CO accumulates over winter to peak in late winter/spring and shows a minimum in late summer. Deviations from this OH-driven seasonal cycle are caused by anomalous sources, and on a large-scale are often due to wildfires(23). The OH-driven spring CO peak and late summer minimum is the seasonal pattern observed in column CO for all regions prior to 2011. In contrast, the recent time period (2012–2018) shows an emerging summer CO peak (in August) during the expected photochemically driven minimum, and the seasonal pattern becomes bimodal."

Reviewer #3 (Remarks to the Author):

General Comments:

The authors have shown increasing wildfires and associated CO emissions in the Pacific Northwest that propagate from west to east with detectable peaks in August in the central and a shift in the peak in the northeastern US. The authors posit that the increased CO concentration down-wind from the origin of the fire along with contemporaneous increases in satellite-derived aerosol optical density suggest that impaired air quality from remote wildfires in the Pacific Northwest could be affecting the health of people in the Central and Northeastern US. The study is further enhanced by associating respiratory mortality to the increase in wildfire emissions in Colorado for 2002-2011 compared to 2012-2018.

Specific Comments:

Pg. 2. L. 20-22. Apart from the climate related effects driving increases in the extent and expansion of

the wildfire season, another factor that is often overlooked is 100 years of land management policy in the U.S. that prioritized suppression of fire and limited prescribed fire. These policies that have resulted in forest overgrowth, susceptibility to disease and insect infestation have resulted in abundant fuel for wildfire and a lack of natural firebreaks. The land management policy contribution to the current wildfire crisis is not the focus of the paper, but to the extent you call for actions to reduce wildfire and protect health those actions will need to include a cohesive strategy for land management.

- Thank you for highlighting that humans impact fire occurrence in a multitude of ways. We have added the following to L18:

"Humans also impact the occurrence of wildfires through land use change (8), increasing ignitions (9), and land management policies such as fire suppression and prescribed burning (10)".

We also considered adding a statement about wildfire management strategies, but ultimately felt that policy prescriptions are beyond the scope of this paper.

Pg. 2. L. 32-33. Consider modifying the following sentence "The influence of PNW fires on other atmospheric trace gas pollutants, as well as the down-wind impacts on air quality, requires further investigation." to "The influence of PNW fires on other atmospheric trace gas and aerosol pollutants, as well as the down-wind impacts on air quality and human health, requires further investigation." In your study CO essentially is serving as a surrogate indicator of other emission pollutants, primarily smoke (PM) that can have human health effects. Introducing aerosols in the previous sentence will strengthen the role of CO as stated in L. 34-36. It's important to get this point made early in the paper as one would not anticipate any health effect of CO at the levels measured. The EPA has established two primary National Ambient Air Quality Standards for carbon monoxide (CO), one averaged over eight-hours (9 ppm not to be exceeded more than once per year) and a one-hour averaging time (35 ppm not to be exceeded more than once per hour). As shown in Figure 2 the peak concentrations of CO from 2002 to 2018 are approximately 120 ppb, a value that is about 75 times less than the standard for the 8-hour average.

- Although the CO might be dangerously high close to the wildfires, Reviewer#3 is correct in that for our study we are interested in CO as a tracer of fire pollution transport, and CO is indicative of other species in the fire plumes such as PM2.5 and ozone that are more relevant for health impacts. We have changed the sentence as suggested (L29-31):

"The influence of PNW wildfires on other atmospheric trace gas and aerosol pollutants, as well as the down-wind impacts on air quality and human health, requires further investigation."

Pg. 5. L. 86-87. What was the reasoning to separate the time series into the two intervals chosen? Based on Pg. 7. L. 132 was the choice based on an empiric observation that the "Inventories also show enhanced August fire CO in the PNW for CO in the PNW for 2012-2018 to 2002-2011?"

- We separate the time series based on the empirical observation of the new August CO peak after 2011 (described L85). We have adjusted the text L94:

"Based on the emergence of the CO peak after 2011, we separate the time series into two time periods, 2002-2011 and 2012-2018, to investigate the average seasonal cycles."

Pg. 6. L. 96-106. The presentation of seasonal cycles of atmospheric CO concentration is clear and well done.

- Thank you for your feedback.

Pg. 7. L. 130. "pf" should read "of".

- Corrected (L139)

Pg. 9. L. 166-168. When describing the health risks associated with wildfire smoke exposure your statements express a high level of certainty in the associations. While this unequivocally true for respiratory disease, the associations with cardiovascular and birth outcomes are as of now not fully defined. Recommend softening the statements on health effects to say something like, "Wildfire smoke increases health risks related to acute respiratory³⁹, ⁴⁰ and evidence is now emerging suggesting an adverse effect on cardiovascular conditions⁴⁰, ⁴¹, and pregnancy outcomes⁴²."

- Thank you, we have changed the sentence L182-184 to:

"Wildfire smoke increases health risks related to acute respiratory conditions (41,42) and evidence is emerging for an adverse effect on cardiovascular conditions, (42,43) and pregnancy outcomes (44)."

Pg. 9. L. 168-169. With respect to the following statement "Although the mortality increases from short-term exposure to ambient PM_{2.5} are well established⁴³, studies specific to short-term wildfire smoke exposure to smoke are limited." consider the following literature:

- **1: Doubleday A, et al. Mortality associated with wildfire smoke exposure in Washington state, 2006-2017: a case-crossover study. Environ Health. 2020 Jan 13;19(1):4. Doi: 10.1186/s12940-020-0559-2.**
- 2: Matz CJ, et al. Health impact analysis of PM_{2.5} from wildfire smoke in Canada (2013-2015, 2017-2018). Sci Total Environ. 2020 Jul 10;725:138506. doi: 10.1016/j.scitotenv.2020.
- **3: Xi Y, Kshirsagar AV, Wade TJ, Richardson DB, Brookhart MA, Wyatt L, Rappold AG. Mortality in US Hemodialysis Patients Following Exposure to Wildfire Smoke. J Am Soc Nephrol. 2020 Aug;31(8):1824-1835. doi: 10.1681/ASN.2019101066.**
- 4: Liu Y, Austin E, Xiang J, Gould T, Larson T, Seto E. Health Impact Assessment of PM_{2.5} attributable mortality from the September 2020 Washington State Wildfire Smoke Episode. medRxiv [Preprint]. 2020 Sep 22:2020.09.19.20197921. doi: 10.1101/2020.09.19.20197921.
- 5: Casey JA, Kioumourtoglou MA, et al. Wildfire particulate matter in Shasta County, California and respiratory and circulatory disease-related emergency department visits and mortality, 2013-2018. Environ Epidemiol. 2020 Dec 21;5(1):e124. doi:10.1097/EE9.
- 6: Haikerwal A, et al. Impact of Fine Particulate Matter (PM_{2.5}) Exposure During Wildfires on Cardiovascular Health Outcomes. J Am Heart Assoc. 2015 Jul 15;4(7):e001653. doi: 10.1161/JAHA.114.001653. PMID: 26178402; PMCID:

- 7: Jones CG, et al. Out-of-Hospital Cardiac Arrests and Wildfire-Related Particulate Matter During 2015-2017 California Wildfires. *J Am Heart Assoc.* 2020 Apr 21;9(8):e014125. doi: 10.1161/JAHA.119.014125.

- Thank you for highlighting these papers. We have considered the suggestion and there are several of these manuscripts that we are reluctant to cite due to either lack of focus on specific wildfire health exposure assessment, or no focus on *mortality* as an outcome. Specifically, the Matz *et al.* citation and Liu *et al.* citations are health impact assessment rather than epidemiology studies and use concentration response functions (CRFs) that are based on general PM2.5 mortality findings rather than wildfire specific CRFs; as background PM2.5 has distinct chemical composition compared to wildfire smoke, we don't think these references are relevant for our paper. The Haikerwal *et al.* paper does not focus on mortality, but other cardiovascular disease events (e.g., out of hospital cardiac arrest, hospitalization and emergency department admissions); likewise, the Jones *et al.* paper only includes out-of-hospital cardiac arrests, not mortality. Xi *et al.* is focused on a very narrow diagnostic group (e.g., patients with end stage renal failure), but did focus on mortality and is already cited (citation 44 in the original, now citation 48 in the revised manuscript). Casey *et al.* did not find evidence of a statistically significant association between mortality and wildfire smoke exposure.

We have added references indicated in bold above and have altered the text on L168-169:

"Although mortality increases from short term exposure to ambient PM2.5 are well-established (45), studies specific to mortality associated with short-term wildfire smoke exposure are limited (46-48)."

Pg. 9. L. 171. The following clause is difficult to understand "... has so far not revealed an effect for general populations downwind of local wildfire events⁴⁵." given that it is among the general population that at-risk individual live and are effected.

- We thank the reviewer for the comment. We have revised the statement L186-189 to the following:

"For individuals with existing co-morbidities (e.g., end stage kidney disease and asthma), wildfire PM_{2.5} has been found to adversely impact mortality (47,48), with limited data of effects for non-accidental mortality downwind of local wildfire events (46,49)."

References:

Akagi, S. K., Yokelson, R. J., Wiedinmyer, C., Alvarado, M. J., Reid, J. S., Karl, T., Crouse, J. D., and Wennberg, P. O.: Emission factors for open and domestic biomass burning for use in atmospheric models, *Atmos. Chem. Phys.*, 11, 4039–4072, doi:10.5194/acp-11-4039-2011, 2011.

Andreae, M. O. and Merlet, P.: Emission from trace gases and aerosols from biomass burning, *Global Biogeochemical Cycles*, 15, 955–966, 2001.

Friedl, M. A., McIver, D. K., Hodges, J. C. F., Zhang, X. Y., Muchoney, D., Strahler, A. H., Woodcock, C. E., Gopal, S., Schneider, A., Cooper, A., Baccini, A., Gao, F., and Schaaf, C.: Global land cover mapping from MODIS: algorithms and early results, *Remote Sens. Environ.*, 83, 287–302, 2002.

Hoelzemann, J. J., Schultz, M. G., Brasseur, G. P., Granier, C., and Simon, M.: Global Wildland Fire Emission Model (GWEM): Evaluating the use of global area burnt satellite data, *J. Geophys. Res.*, 109, D14S04, doi:10.1029/2003JD003666, 2004.

Ito, A. and Penner, J. E.: Global estimates of biomass burning emissions based on satellite imagery for the year 2000, *J. Geophys. Res.*, 109, D14S05, doi:10.1029/2003JD004423, 2004.

van der Werf, G. R., Randerson, J. T., Giglio, L., Collatz, G. J., Mu, M., Kasibhatla, P. S., Morton, D. C., DeFries, R. S., Jin, Y., and van Leeuwen, T. T.: Global fire emissions and the contribution of deforestation, savanna, forest, agricultural, and peat fires (1997–2009), *Atmos. Chem. Phys.*, 10, 11707–11735, doi:10.5194/acp-10-11707-2010, 2010.

REVIEWER COMMENTS

Reviewer #1 (Remarks to the Author):

I am happy with the author's responses. Unless others find significant issues that still need to be addressed, I say to proceed publication.

Reviewer #2 (Remarks to the Author):

Many thanks to the authors for their very thoughtful consideration and detailed responses to my comments. The specific responses, and alterations to the manuscript, are clear and comprehensive and I believe that they increase the relevance of the manuscript, which was already very good. I have no further comments to add and recommend that the manuscript can now be published.

Reviewer #3 (Remarks to the Author):

The reviewer appreciates the responses of the authors to the previous review. One issue remains and that is the final sentence in the abstract. Given the context, the statement "These seasonal pattern changes extend over large regions of North America,17 to the Central USA and Northeast North America regions, potentially impacting the health 18 of millions of people." implies that rising CO potentially has health effects. Don't the authors intend to imply that CO serves as a surrogate or marker of other pollutants emitted from wildfires PM and organics that could have health effects. The estimated ground level CO concentrations would not be expected to produce health effects. The EPA National Ambient Air Quality 8 hour standard is 9ppm, whereas the concentrations described here are 10 or more times lower. Please clarify this point.

We appreciate the time all Reviewers have taken in reading over our responses and the manuscript a second time. Below we address Reviewer comments noted in blue with our responses in black.

Reviewer #1 (Remarks to the Author):

I am happy with the author's responses. Unless others find significant issues that still need to be addressed, I say to proceed publication.

Thank you for your second assessment.

Reviewer #2 (Remarks to the Author):

Many thanks to the authors for their very thoughtful consideration and detailed responses to my comments. The specific responses, and alterations to the manuscript, are clear and comprehensive and I believe that they increase the relevance of the manuscript, which was already very good. I have no further comments to add and recommend that the manuscript can now be published.

Thank you for your second assessment.

Reviewer #3 (Remarks to the Author):

The reviewer appreciates the responses of the authors to the previous review. One issue remains and that is the final sentence in the abstract. Given the context, the statement "These seasonal pattern changes extend over large regions of North America,17 to the Central USA and Northeast North America regions, potentially impacting the health 18 of millions of people." implies that rising CO potentially has health effects. Don't the authors intend to imply that CO serves as a surrogate or marker of other pollutants emitted from wildfires PM and organics that could have health effects. The estimated ground level CO concentrations would not be expected to produce health effects. The EPA National Ambient Air Quality 8 hour standard is 9ppm, whereas the concentrations described here are 10 or more times lower. Please clarify this point.

Thank you for highlighting this potential for miscommunication. We have changed the concluding abstract sentence (L11-13) from:

These seasonal pattern changes extend over large regions of North America, to the Central USA and Northeast North America regions, potentially impacting the health of millions of people.

to:

These seasonal pattern changes extend over large regions of North America, to the Central USA and Northeast North America regions, **indicating that transported wildfire pollution could potentially impact** the health of millions of people.

Additional changes:

We have edited some of the methodology and statistical results based on a request from the editorial team, including two new tables summarizing statistics.